# A hydro-osmotic coarsening theory of biological cavity formation

**Mathieu Le Verge-Serandour** [iD]**, Hervé Turlier** [iD] *

Center for Interdisciplinary Research in Biology, Collège de France, PSL Research University, CNRS UMR7241, INSERM U1050, Paris, France

* herve.turlier@college-de-france.fr

## Abstract

Fluid-filled biological cavities are ubiquitous, but their collective dynamics has remained largely unexplored from a physical perspective. Based on experimental observations in early embryos, we propose a model where a cavity forms through the coarsening of myriad of pressurized micrometric lumens, that interact by ion and fluid exchanges through the intercellular space. Performing extensive numerical simulations, we find that hydraulic fluxes lead to a self-similar coarsening of lumens in time, characterized by a robust dynamic scaling exponent. The collective dynamics is primarily controlled by hydraulic fluxes, which stem from lumen pressures differences and are dampened by water permeation through the membrane. Passive osmotic heterogeneities play, on the contrary, a minor role on cavity formation but active ion pumping can largely modify the coarsening dynamics: it prevents the lumen network from a collective collapse and gives rise to a novel coalescence-dominated regime exhibiting a distinct scaling law. Interestingly, we prove numerically that spatially biasing ion pumping may be sufficient to position the cavity, suggesting a novel mode of symmetry breaking to control tissue patterning. Providing generic testable predictions, our model forms a comprehensive theoretical basis for hydro-osmotic interaction between biological cavities, that shall find wide applications in embryo and tissue morphogenesis.

## Author summary

The formation of a single biological cavity, or lumen, in tissues and embryos has been widely studied experimentally but the collective dynamics of multiple lumens has received much less attention. Here, we focus on a particular type of lumens, which are located at the adhesive side of cells and can therefore interact directly through the intercellular space, as recently observed in the very first stages of embryogenesis. We propose a generic model to describe the hydraulic and osmotic exchanges between lumens themselves, and with the surrounding cellular medium. Lumens are pressurized by a surface tension, which leads naturally to their coarsening into a single final cavity through hydraulic exchanges. With extensive numerical simulations and mean-field theory we predict that such coarsening dynamics follows a robust scaling law, that barely depends on concentration heterogeneities between lumens. On the contrary, active osmotic pumping largely

**Data Availability Statement:** There are no primary data in the paper; the simulation code and Python notebooks for data analysis are available at https://github.com/VirtualEmbryo/hydroosmotic_chain

and we have archived them on Zenodo (DOI: 10. 5281/zenodo.5108774).

**Funding:** The research was supported by the Fondation Bettencourt-Schueller, the CNRS ATIP-Avenir program, and the College de France. HT received his salary from CNRS and MLVS received his salary from from the Fondation Bettencourt-Schueller. The funders had no role in study design, data collection and analysis, decision to publish, or preparation of the manuscript.

**Competing interests:** The authors have declared that no competing interests exist.

influences the collective dynamics by favoring lumen coalescence and by biasing the position of the final cavity. Our theoretical work highlights the essential role of hydraulic and osmotic flows in morphogenesis.

## Introduction

Biological cavities—also known as lumens—are ubiquitous in tissues and embryos. The successful opening and shaping of cavities of various topologies (tubes, network of cavities. . .) during development is essential to organ functions in adult organisms. Lumens appear generally at the non-adhesive side of polarized cells and are called apical [1], to distinguish them from adhesive contacts of polarized cells called basolateral. Common mechanisms for the formation of apical lumens have been described in various model systems, and include: apical membrane secretion, cavitation by cell apoptosis and paracellular ion transport [2, 3]. But the emergence of basolateral cavities, such as blastocoel cavities in early embryos or some cysts in tissues, has received much less attention. Apical lumens are generally sealed at their borders by tight junction proteins [4], which hinder paracellular water leakage [5]. On the contrary, basolateral cavities are not sealed and can directly communicate through the intercellular space, where pressurized fluid may flow and concentrated ions or osmolytes may diffuse. This theoretical manuscript aims therefore at describing primarily the coupled dynamics of basolateral lumens through water and ion exchanges.

In vitro, MDCK monolayers spontaneously generate fluid-filled domes (also called blisters) at their basal surface, when they are plated on a 2D glass surface [6, 7], and cysts in 3D when they are maintained in suspension [8, 9]. This process is characterized by active ion pumping toward the basal side of cells, triggering a water flux in the apico-basal direction [10]. In vivo, the emergence of the blastocoel, the first embryonic cavity, is classically described as the outcome of cell cleavage patterning, where successive divisions take place along the surface plane of the embryo to direct the organization of blastomeres in a spherical sheet [11], such as in cnidarians or echinoderms [12, 13]. In this scenario, supported by numerical simulations [14, 15], the first cleavages leave an initial intercellular space between loosely adhering blastomeres, that is amplified in volume by consecutive planary-oriented divisions, requiring the free movement of water through the nascent epithelial layer to fill the central cavity. But in various vertebrate embryos such as mammals (eutherians) [16], and amphibian embryos [17], the outer epithelial layer may be already sealed by tight junctions before blastocoel formation, making the emergence of a cavity by cleavages alone unclear [18]. Recently, we uncovered an alternative mechanism in mouse embryos [19], whereby adhesive contacts are hydraulically fractured into myriad of microcavities (also called microlumens) that progressively coarsen into a single cavity through fluid exchanges within the $\sim 50$nm wide intercellular space [20, 21] (Fig 1). Yet a detailed physical model for this process, including ion pumping and osmotic exchanges is missing. In fact, similar micron-sized fluid-filled compartments had already been observed at the cell-cell contacts of several early embryos during blastula formation: in mammals [22, 23], in amphibians [24–26] and even in *C. elegans*, although for the latter the resulting cavity rapidly disappears [27]. The transient coarsening phase may have been missed in other species having a tightly compacted morula prior to blastula stage, because of the need to image section planes of the embryo with a high spatiotemporal resolution. In this alternative mechanism, the fluid is driven into the intercellular space thanks to an osmotic gradient, highlighting an evident coupling between hydraulic and osmotic effects. The requirement of ion pumping was

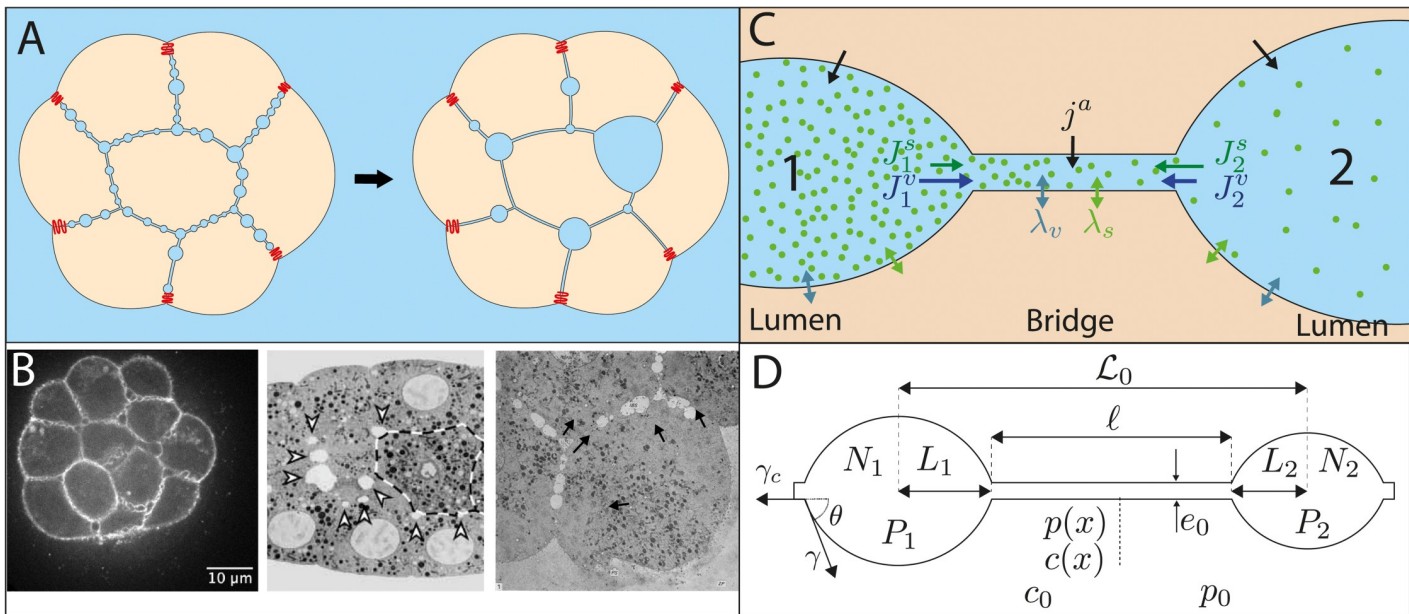

**Fig 1. Network of microlumens in embryos and description of the model.** (A) Schematic of the microlumen coarsening process in a cell aggregate or embryo (tight junctions are marked in red) (B) Examples of microlumen networks in various early embryos. Left: fluorescence microscopy confocal section of a mouse embryo at 32-cell stage [19]. Center: Electron micrograph of a 28-cell *C. elegans* embryo [27]. Right: Electron micrograph of a 9-cell rhesus monkey embryo [23]. (C) Lumens are modeled as distinct compartments interconnected through an intercellular bridge. They can exchange water (*v*) and solutes (*s*) with each others through the bridge (fluxes $J_i^v$ and $J_i^s$) and with surrounding cells by active pumping of rate $j^a$ and through passive permeation of coefficients $\lambda_v$, $\lambda_s$. (D) Parametrization of the two-lumen system: each lumen is characterized by its half-length $L_i \propto \sqrt{A_i}$ (where $A_i$ is the lumen area), its number of moles $N_i$ (leading to a homogeneous concentration $C_i = \frac{N_i}{A_i}$) and its pressure $P_i - p_0 \propto \gamma/L_i$. The intercellular bridge is characterized by its length $\ell(t)$, its thickness $e_0$, and its profiles of solute concentration $c(x)$ and hydrostatic pressure $p(x)$ along its length. The system is supposed embedded in an infinite *cellular* medium, playing the role of chemiostat of concentration $c_0$ and barostat of pressure $p_0$.

indeed already well documented in mammalian embryos [28–32], and some evidence of osmotic control of cavity formation in other embryo types was already put forward earlier [33–35]. Yet, little is known about the hydro-osmotic control of biological cavity formation from a physics perspective [18].

Hydraulic and osmotic flows are indeed more and more recognized as essential determinants of embryo and tissue shaping [19, 36–41]. However only a few physical models describe the interplay between cell mechanics, osmotic effects and fluid flow in morphogenesis [42–45], and in all those previous models, osmolarity is considered spatially homogeneous. Here, we propose a generic physical model to describe the hydro-osmotic coupling between pressurized compartments, explicitly accounting for osmotic gradients and ion pumping. Our model may therefore also form a relevant theoretical framework to describe the coupled dynamics between nurse cells and the future oocyte during Drosophila oogenesis [37, 40], or between germ cells in the nematode *C. elegans* [41]. Here, we do not study however the initial nucleation of small cavities. In embryos, this nucleation process likely involves the hydraulic fracturation of cell-cell contacts, that is beyond the scope of this manuscript [46]. We start, on the contrary, from a preformed and contiguous network of microlumens (Fig 1A and 1B) and we focus on their collective dynamics using numerical simulations and mean-field theory. For the sake of simplicity, we also neglect ion charges and membrane potentials and consider a single solute only. This follows similar hypotheses as a recent model for the growth of a single apical lumen [36].

## Model and results

### Coupled dynamics for two lumens

In this first section, we focus on a 2-lumen system to introduce the fundamental dynamic equations and to characterize the competition between hydraulic and osmotic effects.

**Dynamical equations.** Dynamical equations for two lumens are derived from conservation equations of mass and solutes. The lumens are represented as 2-dimensional compartments connected through a 1-dimensional bridge of width $e_0$ and length $\ell$ along the x-direction (Fig 1D). In the next, variables associated with lumens are denoted by capital letters, while lower-case variables are used for bridges. Each lumen $i$ = 1, 2 is described by its area $A_i$, its hydrostatic pressure $P_i$ and its concentration $C_i = \frac{N_i}{A_i}$, supposed homogeneous, where $N_i$ is the number of solutes (in moles). A lumen is pressurized by a tension $\gamma$ supposed constant, leading to a Laplace's pressure jump $\delta P_i = P_i - p_0 = \gamma \sin\theta / \sqrt{\mu A_i}$ across the membrane, where $\mu = \sin^2\theta/(2\theta - \sin(2\theta))$ is a constant that depends on the contact angle $\theta = \arccos(\gamma_c/2\gamma)$ between the lumens and the bridge with $\gamma_c$ the adhesive tension in the bridge (see S1 Appendix and Fig 1D). The lumens and the bridge are delimited by two lateral membranes of water and solutes permeability coefficients $\lambda_v$ and $\lambda_s$ respectively and are embedded in a an infinite "cellular" medium acting as a chemostat of concentration $c_0$ and as a barostat of pressure $p_0$ (Fig 1D). The mass in a lumen $i$ is modified through a longitudinal hydraulic flux $J_i^v$ toward the bridge and through lateral osmotic fluxes from the cellular medium, that stem from the imbalance between osmotic and hydraulic pressure across membranes (Fig 1C). For practicality, we write mass balance as function of the lumen half-length $L_i = \sqrt{\mu A_i}$

$$\frac{dL_i}{dt} = 2\mu v \lambda_v [\mathcal{R}T\delta C_i - \delta P_i] - \frac{\mu}{2L_i}J_i^v, \tag{1}$$

where $\mathcal{R}$ is the gas constant, $T$ the temperature, $\delta C_i = C_i - c_0$ is the concentration jump across the membranes, and $v = \theta/\sin\theta$ is a geometrical constant function of $\theta$ (see S1 Appendix). The hydraulic flux $J_i^v$ is calculated from boundary conditions in the bridge. The number of moles of solutes in each lumen $N_i$ similarly varies through an outgoing flux $J_i^s$ toward the bridge and through lateral passive fluxes, triggered by the chemical potential difference of the solute across membranes. Adding an active pumping rate $j^a$ by unit membrane length, solute balance reads

$$\frac{dN_i}{dt} = 2vL_i\left[\lambda_s \mathcal{R}T \log\left(\frac{c_0}{C_j}\right) + j^a\right] - J_i^s \tag{2}$$

Conservation equations in the bridge are written along its length $\ell(t)$ in the x-direction. Local mass balance is obtained by calculating incoming and outgoing longitudinal fluxes along the x-axis and lateral osmotic fluxes across the membranes. The longitudinal flux is given by the 2D Poiseuille's flow $q(x) = -\kappa_v \frac{\partial \delta p}{\delta x}$, where $\kappa_v \approx \frac{e_0^3}{12\eta}$ is the hydraulic conductivity, inversely proportional to the solvent viscosity $\eta$. Considering a passive lateral flux of solvent along the two membrane sides, like for the lumens, yields

$$\kappa_v \frac{\partial^2 \delta p}{\partial x^2} + 2\lambda_v[\mathcal{R}T\delta c(x) - \delta p(x)] = 0 \tag{3}$$

where $\delta c(x) = c(x) - c_0$ and $\delta p(x) = p(x) - p_0$ are local concentration and pressure jumps across the membrane. The balance of solutes along the bridge is similarly given by the sum of a longitudinal diffusive flux $e_0 j_d(x) = -e_0 D \frac{\partial \delta c}{\partial x}$ (Fick's law with diffusion coefficient $D$) and passive lateral fluxes of solutes across the two membrane sides, driven by the imbalance of chemical

potential for the solute. Adding an active pumping rate of solutes $j^a$ yields

$$e_0 D \frac{\partial^2 \delta c}{\partial x^2} = 2\mathcal{R}T\lambda_s \log\left(1 + \frac{\delta c}{c_0}\right) - 2j^a \tag{4}$$

where we assumed that the relaxation of concentration in the bridge is very fast compared to other timescales in the problem: $\frac{\partial c}{\partial t} \sim 0$. We also neglect the advection of solutes by hydraulic flows in the bridge (see S1 Appendix Eq (9)). Similarly to [36], solvent and solute exchanges along the bridge are found to be controlled by two characteristic length scales

$$\xi_v = \sqrt{\frac{\kappa_v}{2\lambda_v}}, \qquad \xi_s = \sqrt{\frac{De_0 c_0}{2\lambda_s \mathcal{R}T}} \tag{5}$$

which measure the relative magnitude of longitudinal and lateral passive fluxes of solvent (v) and solutes (s) respectively. They play the roles of screening lengths: a pressure (resp. concentration) gradient along the bridge will be screened on a length larger than $\xi_v$ (resp. $\xi_s$). Therefore, for $\ell \gg \xi_v$ (resp. $\ell \gg \xi_s$), we expect the lumens to become hydraulically (resp. osmotically) uncoupled. On the contrary, for $\ell \ll \xi_{v,s}$, pressure or concentration differences along the bridge may trigger solvent or solute fluxes from one lumen to the other (see S1 Video). In the rest of the manuscript, we will denote by screening numbers the dimensionless ratios $\chi_{s,v} \equiv \xi_{s,v}/\ell_0$, where absolute screening lengths $\xi_{s,v}$ have been rescaled by the mean initial bridge length $\ell_0 = \langle \ell(0) \rangle$. Interestingly, we note that these ratios $\xi_{v,s}/\ell(t)$ count indeed the average number of neighbors that a given lumen can interact with hydraulically (resp. osmotically). We use these ratios as global parameters to characterize screening effects in our system, but it should be noted that the actual pressure or concentration screening numbers vary with the length of the bridge $\ell(t)$.

To simplify our equations, we remark that the typical hydrostatic pressure jump across the plasma membrane of embryonic cells $\delta P_i \sim \delta p \sim 10^{2-3}$Pa is a few orders of magnitude lower than the absolute value of osmotic pressure $\Pi_0 = \mathcal{R}Tc_0 \sim 10^5$ Pa. This implies generically that $\delta C_i, \delta c \ll c_0$ (see S1 Appendix) and allows us to derive analytic solutions for solvent and solute fluxes $\pm q(x)$ and $\pm e_0 j_d(x)$ from the bridge Eqs (3) and (4). By equating these fluxes, evaluated at bridge boundaries $x = \mp\ell(t)/2$ with outgoing fluxes from the lumens $J_i^v$, $J_i^s$ for $i = 1, 2$, and by fixing the system size $\mathcal{L}_0 = L_1(t) + L_2(t) + \ell(t) = $ cte, we close our system of equations (see S1 Appendix). It is important to remark that the previous geometric relation between lumens half-lengths supposes that the center of mass of each lumen is fixed in space. While, in all generality, asymmetric fluxes between the two ends of a given lumen may lead to the slow motion of its center of mass, in practice we expect such movements to be hampered by the adhesive friction created by cadherin molecules in bridges. In the context of mouse embryo blastocoel formation, no measurable movements of microlumens was indeed observed over the course of the coarsening process [19]. Furthermore, from a theoretical perspective, it was proved earlier that such motions have generically a minor effect on the coarsening behavior [47].

Denoting $L_0$ and $N_0$ the mean initial lumen size and solute number, we now introduce dimensionless variables $L_i = \bar{L}_i L_0$, $\ell = \bar{\ell}\ell_0$, $N_i = \bar{N}_i N_0$, $j^a = \bar{j}^a \lambda_s \mathcal{R}T$, $J_i^v = \bar{J}_i^v \frac{L_0^2}{\tau_v}$, $\bar{J}_i^s = \bar{J}_i^s \frac{N_0}{\tau_s}$, and dimensionless parameters $N_0 = c_0 L_0^2$ and $\epsilon = \frac{\gamma \sin \theta}{L_0 \Pi_0}$. The dynamics of the system reduces finally to four coupled ordinary differential equations (ODEs) for $\bar{N}_i$ and $\bar{L}_i$ ($i = 1, 2$):

$$\frac{d\bar{L}_i}{d\bar{t}} = \mu v \left[ \mu \frac{\bar{N}_i}{\bar{L}_i^2} - 1 - \frac{\epsilon}{\bar{L}_i} \right] - \frac{\mu}{2\bar{L}_i} \bar{J}_i^v \tag{6}$$

$$\frac{\tau_s}{\tau_v}\frac{d\bar{N}_i}{d\bar{t}} = 2v\bar{L}_i\left[1 - \mu\frac{\bar{N}_i}{\bar{L}_i^2} + \bar{j}^a\right] - \bar{J}_i^s \tag{7}$$

$$1 = \bar{L}_1 + \bar{L}_2 + \bar{\ell} \tag{8}$$

where outgoing fluxes $\bar{J}_i^{v,s} = f_i^{v,s}(\bar{N}_{1,2}, \bar{L}_{1,2}, \bar{\ell})$ are functions coupling dynamically the two lumens $i = 1, 2$. The dynamics is controlled by the following solvent and solute equilibration timescales

$$\tau_v = \frac{L_0}{2\lambda_v\Pi_0} , \tau_s = \frac{L_0 c_0}{2\lambda_s\mathcal{R}T} \tag{9}$$

**Hydro-osmotic control of coarsening.** To characterize the competition between osmotic and hydraulic effects, we focus first on the factors controlling the instantaneous net solvent flow between two lumens. Any asymmetry in size or tension between two lumens is expected to generate a hydrostatic pressure difference $\Delta P = P_2 - P_1$, triggering a net solvent flow $J_{2\to1}^v = J_2^v - J_1^v$. But the bridge may also act as an (imperfect) semi-permeable membrane, that can carry osmotically-driven solvent flows if a concentration asymmetry $\Delta C = C_2 - C_1$ exists between the lumens. These two kinds of solvent flows, hydraulic and osmotic, may enter in competition if $\Delta C\Delta P > 0$, as shown in Fig 2A where we plotted the signed value of the net dimensionless flow $\bar{J}_{2\to1}^v$ as function of dimensionless concentration and pressure asymmetries $\Delta_C = (C_2 - C_1)/(C_2 + C_1)$ and $\Delta_P = (P_2 - P_1)/(P_2 + P_1)$. For a given set of screening numbers $\chi_v$, $\chi_s \sim 1$, the net solvent flow direction expected from Laplace's pressure (from small to large lumens) may be outcompeted by an osmotic flow in the reverse direction if a sufficiently large

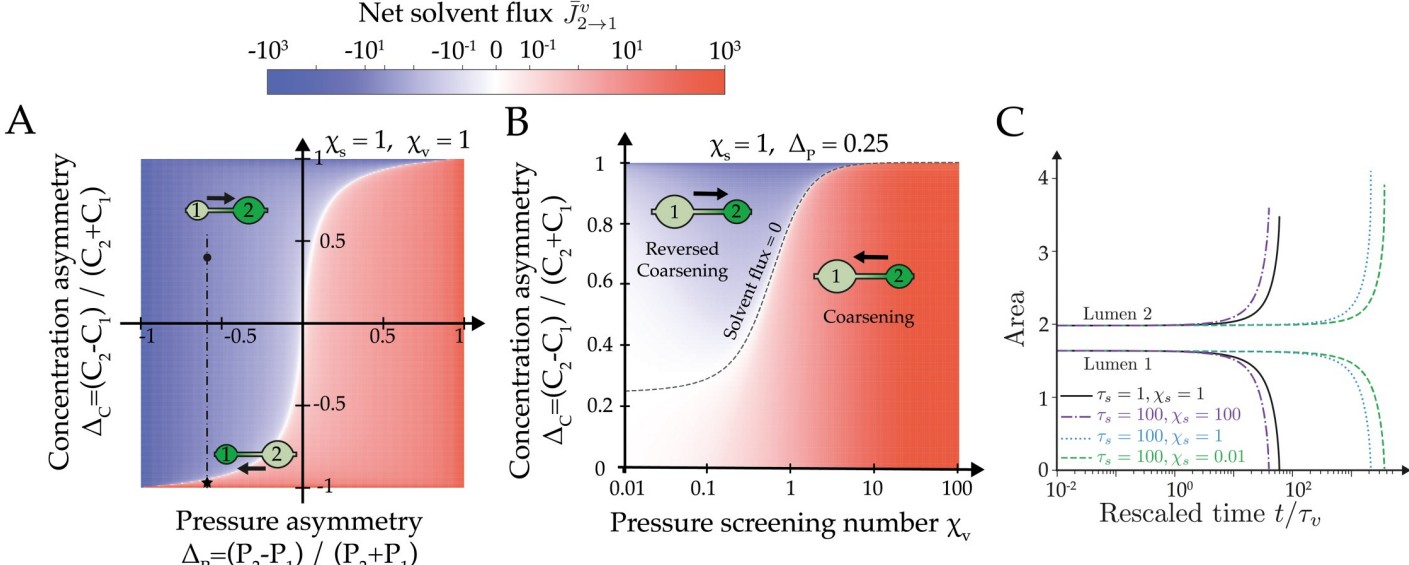

**Fig 2. Two-lumen dynamics without active pumping.** (A) Diagram of the net flow $\bar{J}_{2\to1}^v = \bar{J}_2^v - \bar{J}_1^v$ as a function of concentration and pressure asymmetries $\Delta_C = (C_2 - C_1)/(C_1 + C_2)$ and $\Delta_P = (P_2 - P_1)/(P_1 + P_2)$, for $\chi_{s,v} = 1$. The flow direction is schematized for two sets of values $(\Delta_P, \Delta_C)$. (B) Diagram of the net flow $\bar{J}_{2\to1}^v$ as function of the concentration asymmetry $\Delta_C$ and pressure screening number $\chi_v$. $\chi_s = 1$ and $\Delta_P = 0.25$. (C) Time evolution of the area of two lumens as function of $\chi_s$ and $\tau_s$. Other simulation parameters are $\chi_v = 10$, $\tau_v = 1s$. In all the figures, $\bar{j}^a = 0$.

concentration asymmetry exists between the two lumens. In practice, we observe that the relative asymmetry has to be much larger for concentrations than for pressures to reverse the coarsening direction. This let us anticipate in general a limited influence of osmotic gradients on lumen coarsening dynamics. As further indication for this trend, we find that the magnitude of the solvent flow depends primarily on the pressure screening number $\chi_v$ (Fig 2B), while the screening of concentration asymmetries has no measurable effect for $\chi_s > 1$ and only a mild influence for $\chi_s < 1$ (see S1 Fig). From a dynamical point of view, osmotic effects could however have a strong influence in the limit $\tau_s \gg \tau_v$ and $\chi_s \ll 1$, where solutes are expected to become almost trapped in lumens, their exchange dynamics with the cellular medium and other lumens being largely slowed down. Indeed, if the size of a given lumen drops rapidly, its solute concentration will rise (see S3 Fig), generating an osmotically-driven solvent flow that may oppose to the pressure-driven flow. This effect, reminiscent of the osmotic stabilization of emulsions and foams by trapped species [48, 49], is illustrated in Fig 2C where we plot the dynamics of two lumens starting at osmotic equilibrium for $\tau_s = 100\tau_v$ and for decreasing values of $\chi_s$ from 100 to $10^{-2}$ ($\chi_v = 10$). In this regime, we observe that the coarsening dynamics is slowed down by a few orders of magnitude through the combined effects of slow solute relaxation and large concentration screening. However, and in contrary to foams or emulsions [48, 49], our system cannot reach complete stabilization, since lumens ultimately homogenize their concentrations by exchanging fluid with surrounding cells. This let us therefore anticipate that lumen coarsening with negligible pumping may always lead either to a single final cavity for low pressure screening, or to the synchronous collapse of all lumens in regime of high pressure screening.

**Influence of active pumping.** Then, we explore the influence of active pumping on the dynamics of two lumens. Ion pumps, such as the Na/K ATPase, can actively transport ions from/to the cell against the chemical potential difference across the membrane by hydrolyzing ATP. Here, we consider an initial size ratio $\bar{L}_2(0)/\bar{L}_1(0) = 1.1$ and a nonzero pumping rate $\bar{j}_1^a$ in the lumen 1 only ($\bar{j}_2^a = 0$). We plot in Fig 3 the outcome (or fate) of the two lumens as function of $\bar{j}_1^a$ and $\chi_v$. We identify four different possible fates (see S1 Video): collapse of the two lumens (triangles), coarsening $1 \rightarrow 2$ (disks), reversed coarsening $2 \rightarrow 1$ (stars) and coalescence (squares). At vanishing pumping $\bar{j}_1^a \sim 0$, both lumens are under tension and may either coarsen if their pressure difference is not screened ($\chi_v \gtrsim 1$) or collapse at large pressure screening ($\chi_v \lesssim 1$). When pumping is increased in the lumen 1, a third regime that we called reversed coarsening appears, where the coarsening direction is inverse to the one expected from the initial size asymmetry: this happens when the growth of the lumen 1 driven by pumping overcomes its discharge into the lumen 2, reversing the flow direction. Above some pumping rate a fourth dynamical regime appears, where both lumens grow faster than they exchange their content, leading to a fast decrease of the bridge length and lumen collision. Upon collision, two lumens are merged: this is referred as coalescence, and may be observed in late stages of mouse blastocoel embryo formation [19] but also in droplet coarsening [50, 51]. As we do not calculate the profile of the interface as in [52] but rather consider an approximate description of the lumens shape, we consider the coalescence events to be instantaneous, such that the total mass of colliding lumens is conserved. We find therefore that active pumping can greatly modify the dynamics of two lumens, which let us anticipate its major role in the coarsening of a network of lumens.

To summarize, the 2-lumen system draws already major trends in lumen coarsening: **(1)** Passive osmotic differences have a limited influence on the coarsening of two lumens, which remains largely dominated by hydraulic fluxes, and hence by hydrostatic pressure differences, apart in the limit case of very slow solute equilibration $\tau_s \gg \tau_v$ and $\chi_s \ll 1$, that leads to a

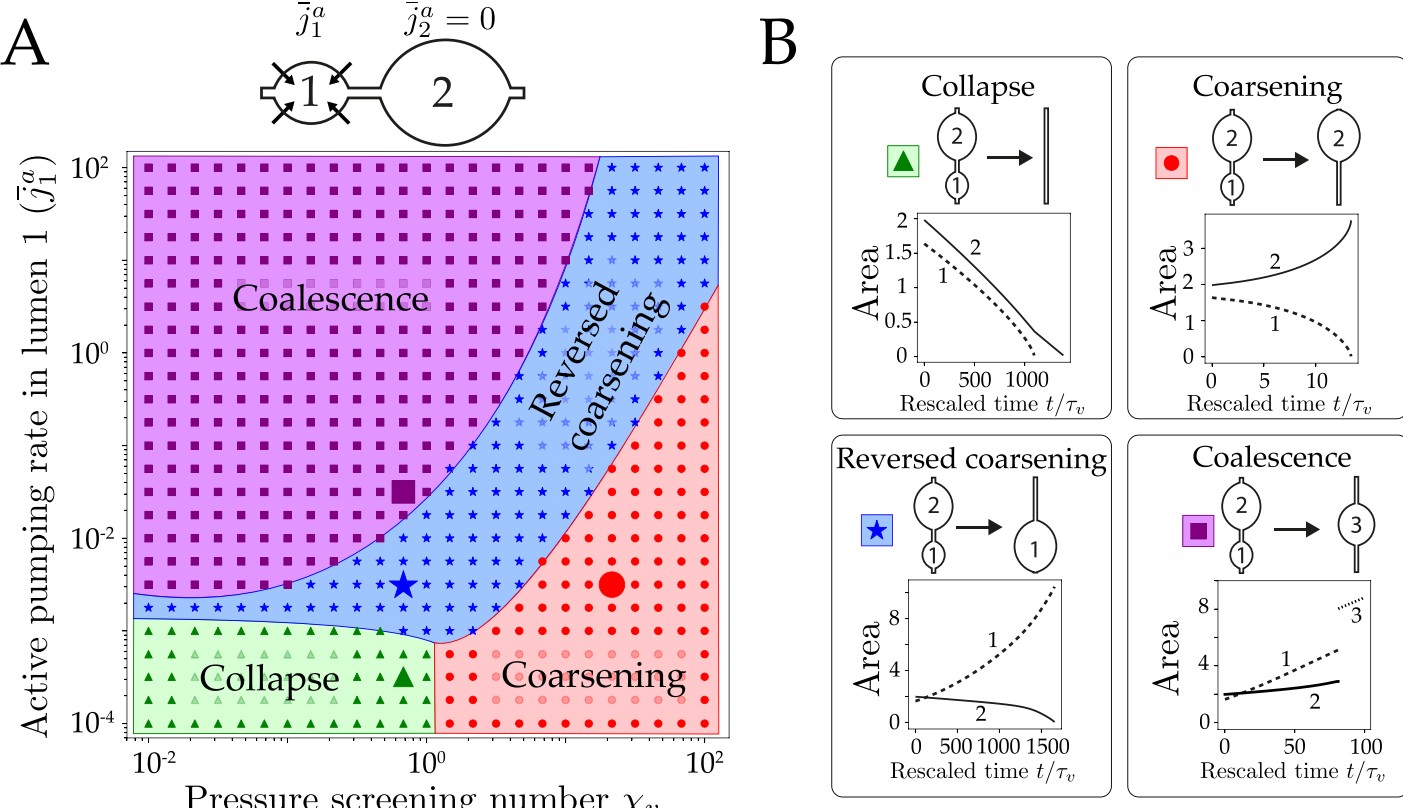

**Fig 3. Classification of two lumens dynamics with active pumping.** (A) Fate diagram for 2 lumens as function of the active pumping $\bar{j}_1^a$ of lumen 1 and pressure screening number $\chi_v$, at fixed $\chi_s = 1$. (B) Typical time evolution of the areas of lumen 1 (dotted line) and lumen 2 (full line) in the four cases. Four behaviors are found numerically: collapse for large screening and low pumping, coarsening and reversed coarsening at intermediate values and coalescence for large pumping. The delineation of the regions have been drawn by hand, for presentation purpose. Both lumens start at osmotic equilibrium ($C \sim c_0$) but with a size asymmetry $\bar{L}_2(0) = 1.1\bar{L}_1(0)$.

notable slowing-down of the overall dynamics. **(2)** Active pumping can largely change the dynamics, by lowering the risk of collective collapse in large pressure screening regimes ($\chi_v \ll 1$), and by driving a novel mode of coarsening by coalescence, when lumen growth outcompetes hydraulic exchanges between lumens. It may also reverse the "natural" direction of coarsening, generally expected from small to large lumens.

## Coarsening dynamics for a 1-dimensional chain

In this section, we turn to the dynamics of a 1-dimensional chain of $\mathcal{N}$ lumens (Fig 4A). Each lumen of label $i$ is described by its position, its size $L_i = \sqrt{\mu A_i}$ and its number of moles of solutes $N_i$, and is connected to up to two neighboring lumens by a bridge $(i, j)$ of length $\ell_{ij}(t)$. The chain has a fixed total length $\mathcal{L}_0 = 2\sum_i L_i(t) + \sum_{\{ij\}} \ell_{ij}(t)$ and its borders are fixed and sealed (these borders may be seen as the apical tight junctions sealing the embryo/tissue at its interface from the external medium). We assume, again, that the center of mass of each lumen remains at a fixed position. These positions are initialized randomly along the chain, and initial chain configurations are generated by picking lumens areas $A_i$ and bridges lengths $\ell_{ij}$ from Gaussian distributions of respective means $A_0 = 1$ and $\ell_0 = 10$, and respective standard deviations 0.2 and 2. All our simulations start at osmotic equilibrium ($\delta C_i \sim \delta c \sim 0$), and unless otherwise specified, we set $\tau_v = \tau_s = 1s$. The coupled ODEs (6) (7) for the lumens $i = 1, \ldots, \mathcal{N}$ are

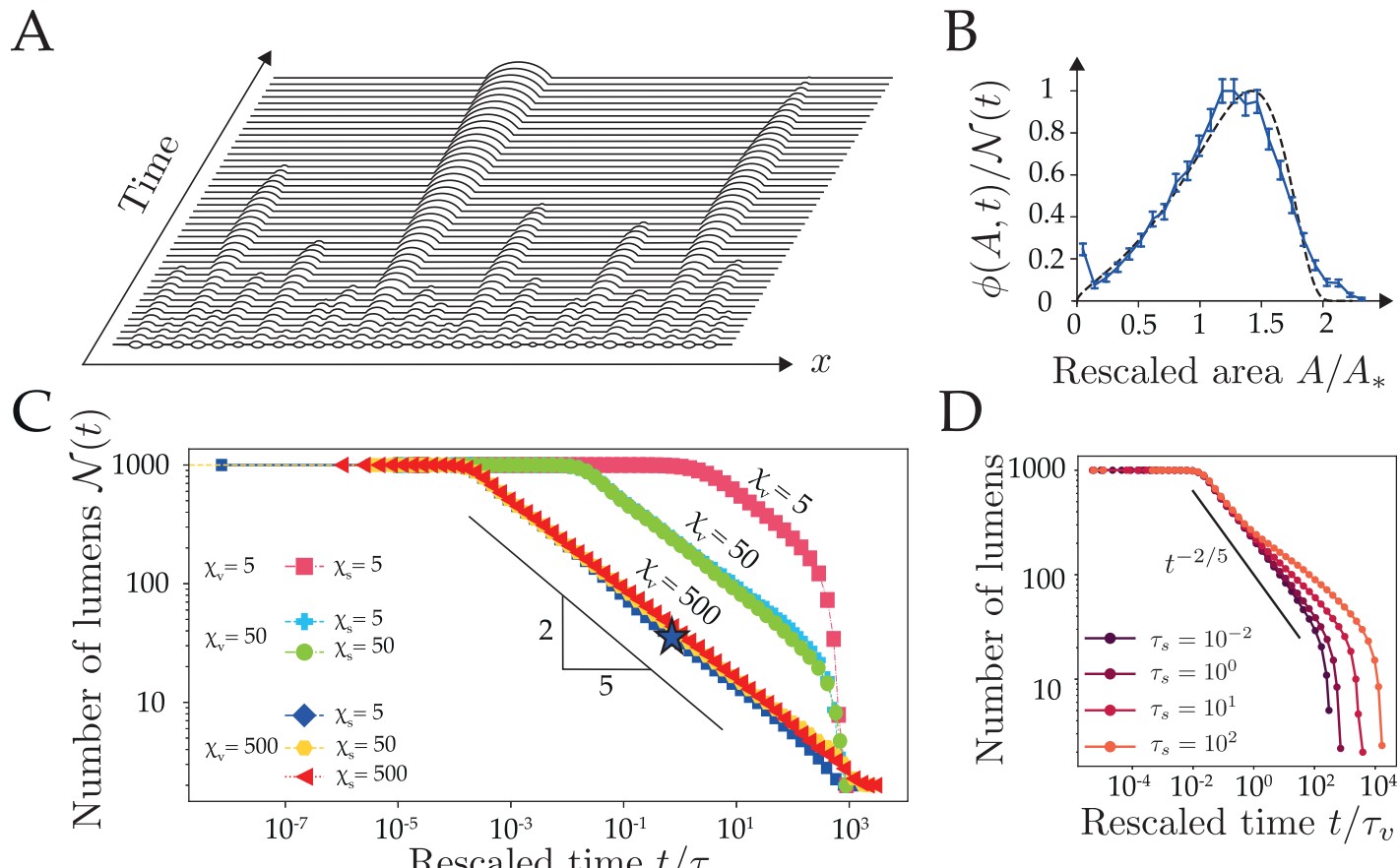

**Fig 4. Collective dynamics of a 1-dimensional chain of lumens.** (A) Typical spatiotemporal hydro-osmotic coarsening dynamics of a 1-dimensional chain. Here $\mathcal{N}(0) = 30$ lumens evolve from osmotic equilibrium with $\chi_v = 100$, $\chi_s = 1$, $\tau_{s,v} = 1s$. (B) Lumen area distribution as function of rescaled area $A/A_*$, where $A_* = \int A\phi(A, t)dA$ is the mean field area with respect to this distribution. The plain curve is an average over 100 simulations with $\chi_v = 500$, $\chi_s = 5$ and taken at the time-point indicated by a star on panel C. The dashed line is the theoretical self-similar distribution given in the S1 Appendix Eq (48). (C) Plot of the number of lumens as function of the rescaled time $t/\tau_v$ for various values of the screening numbers $\chi_{v,s}$ where each simulation starts at $\mathcal{N}(0) = 1000$ lumens. The scaling law $t^{-2/5}$ (black line) is shown as a guide for the eye. Relaxation times $\tau_{s,v} = 1s$. (D) Plot of the number of lumens as function of the rescaled time $t/\tau_v$ for $\tau_v = 1s$, $\chi_v = 50$, $\chi_s = 5$ and increasing values of the solute relaxation time $\tau_s$ from $10^{-2}$ s to $10^2$ s. A deviation from the scaling law $t^{-2/5}$ (plotted in plain line as a reference) is observed for large $\tau_s$, indicative of an osmotic stabilization effect, which slows down the coarsening. Each curve in panels C and D, is an average of 20 simulations.

integrated numerically leading to the typical lumen dynamics shown in Fig 4A and S2 Video. After a few steps, some lumens may collapse or enter in collision; when such topological events happen, the integration is stopped, the configuration of the chain is recalculated to remove or merge implicated lumens before restarting the integration (see Methods & materials and S1 Appendix).

**Coarsening without pumping.** To characterize the average dynamics of a chain, we plot in Fig 4C the number of lumens as function of time for various values of the initial screening numbers $\chi_v$ and $\chi_s$. After a plateau, we generically observe a coarsening behavior characterized by a dynamic scaling law $\mathcal{N}(t) \propto t^{-\frac{2}{5}}$. This scaling exponent is in fact reminiscent of the dynamics of a purely hydraulic chain of droplets, as studied in thin dewetting films [47, 53, 54]. In the limit of small $\chi_v$, we observe yet a rapid collapse of the chain, indicative of the overall uncoupling of lumens, which loose passively their content by permeation through the membrane, as observed for the 2-lumen system (Fig 3A). As far as lumens remain hydraulically coupled, the coarsening is self-similar in time with a scaling exponent that is largely unaffected

by pressure and concentration screening numbers (Fig 4C). The main difference with a pure hydraulic dynamics can be observed in the onset of coarsening, that is characterized by a typical timescale $T_h = \frac{2\tau_v \ell_0 L_0}{\mu \epsilon \xi_v^2} = \frac{2\ell_0 L_0^3}{\mu \sin\theta \, \gamma \kappa_v}$ that increases with a decreasing pressure screening length $\xi_v$. To understand the link between our dynamics and a pure hydraulic system, we consider the limit $\chi_v \gg 1$, $\chi_s \ll 1$, and $\tau_s \ll \tau_v$, where concentrations relax rapidly and lumens are fully coupled hydraulically. In this limit, our system of equations reduces to a single dynamical equation where the contribution of solutes has vanished. It becomes analytically equivalent to the mean-field model proposed in [52, 55] to describe the coarsening of thin dewetting films:

$$\frac{d\bar{L}_i}{dt} = \frac{1}{T_h \bar{\ell}_{ij} \bar{L}_i} \left( \frac{1}{\bar{L}_j} - \frac{1}{\bar{L}_i} \right) \tag{10}$$

We will refer to this limit as the *hydraulic chain*, where osmotic effects have disappeared. In such limit, the scaling law for the number of lumens $\mathcal{N}(t) \sim t^{-2/5}$ as well as the distribution of lumens size $\phi(A, t)$ can be predicted from a mean-field theory of Lifshitz-Slyosov-Wagner type (see S1 Appendix Eqs (47) and (48)) [52, 55, 56]. In Fig 4B, we compare the size distributions of lumens predicted by the hydraulic chain mean field theory and obtained from our simulations for $\chi_v = 500$, $\chi_s = 5$. The very close agreement indicates that the hydraulic chain limit is generically a good approximation for the behavior of our hydroosmotic chain. To challenge the validity of the hydraulic chain approximation, we chose parameters $\tau_s \gg \tau_v$ and $\chi_s \ll 1$ favoring the retention of solutes in lumens, as we did for the 2-lumen system in Fig 2C. We plot on Fig 4D the time evolution of the chain for increasingly lower solute equilibration times (increasing $\tau_s$) and observe indeed a progressive deviation from the scaling law $t^{-2/5}$ when $\tau_s \gg \tau_v$, indicative of a slowing-down of coarsening. Like for the two-lumen system, solutes are transiently trapped within lumens, which triggers larger osmotic asymmetries between lumens (S3 Fig) that may compete with pressure-difference driven flows and slow down the hydraulic coarsening of the chain (Fig 4D).

**Coarsening with active pumping.** We finally study the influence of active pumping on the collective chain dynamics. We first assume a homogeneous pumping rate $\bar{j}^a$ along the chain. In Fig 5A we plot the time evolution of the number of lumens $\mathcal{N}$ as function of $t/T_h$ for increasing values of the pumping rate $\bar{j}^a$ from $10^{-2}$ to 10 ($\chi_v = 50$, $\chi_s = 50$). The dynamics is now typically characterized by two coarsening phases that are separated by a novel timescale $T_p = \frac{\tau_v}{\mu\nu\,\bar{j}^a} = \frac{\lambda_s L_0 \mathcal{R}T}{2\lambda_v \Pi_0 \bar{j}^a}$. This timescale measures the competition between active solute pumping, controlled by $\bar{j}^a$, and passive solute relaxation, controlled by $\lambda_s$, in triggering solvent permeation, limited by $\lambda_v$. For $t \ll T_p$ we recover the hydraulic chain limit characterized by a power-law $t^{-2/5}$ as previously. For $t \gg T_p$, the number of lumens first stabilizes into a plateau before decreasing again with a novel scaling law $\mathcal{N}(t) \sim t^{-1}$. This novel coarsening regime is dominated by lumen coalescence (see S3 Video) and may be understood with the following arguments. Considering the previous hydraulic chain limit ($\chi_v \gg 1$, $\chi_s \ll 1$ and $\tau_s \ll \tau_v$) with nonzero $\bar{j}^a$, we can derive a modified expression for lumen dynamics (see S1 Appendix Eq (38))

$$\frac{d\bar{L}_i}{dt} = \frac{1}{T_h \bar{\ell}_{ij} L_i} \left[ \frac{1}{\bar{L}_j} - \frac{1}{\bar{L}_i} \right] + \frac{1}{T_p} \tag{11}$$

In this limit, the last term in Eq (11) is inversely proportional to $T_p$ and corresponds exactly to the rate of osmotic lumen growth triggered by active pumping. Taking the limit $t \gg T_p$, the

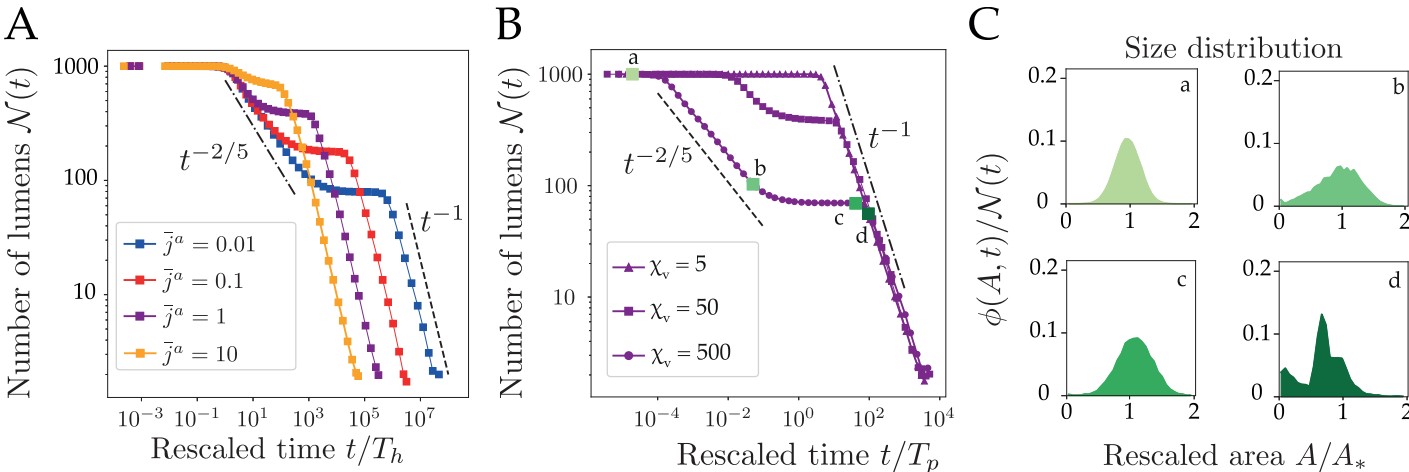

**Fig 5. Collective dynamics of a 1-dimensional chain of lumens with active pumping.** (A) Coarsening of 1-dimensional chains with uniform active pumping $\bar{j}^a$: plot of the number of lumens as function of the rescaled time $t/T_h$ for increasing values of the pumping rate $\bar{j}^a = 10^{-2}, 10^{-1}, 1$ and 10. Screening numbers are $\chi_v = 50, \chi_s = 5$. (B) Number of lumens as function of the rescaled time $t/T_p$ for three values of the pressure screening number $\chi_v$, for $\bar{j}^a = 1$ and $\chi_s = 5$. Each curve in panels A and B is an average of 20 simulations; relaxation times $\tau_{s,v} = 1s$. (C) Area distribution $\phi(A, t)$ at four different times indicated on the plot in (B), with 100 simulations.

Eq (11) becomes $\frac{dL_i}{dt} \simeq \frac{1}{T_p}$, and since lumens are assumed not to move, such that the distance between their center of mass $\bar{L}_i(t) + \bar{L}_j(t) + \bar{\ell}_{ij}(t)$ remains constant, we deduce $\frac{d\bar{\ell}_{ij}}{dt} \simeq -\frac{2}{T_p}$. The average length of bridges decreases therefore linearly in time, such that the rate of coalescence will naturally scale as $\mathcal{N}(t) \sim (t/T_p)^{-1}$, as shown on Fig 5A and 5B. One may note that this scaling exponent shall remain independent of the dimension of the system, as far as the growth is proportional to the extent of lumen boundaries, which forms an hypersurface in the ambient space (see S1 Appendix Eq (50)). For large enough pumping rate $\bar{j}^a$, we reach a limit where $T_p \sim T_h$, such that the hydraulic coarsening phase disappears, leaving only a coarsening dynamics dominated by coalescence, as we observe on Fig 5A and 5B and S3 Video. On Fig 5C we plot the size distribution of lumens at various timepoints of a dynamics where hydraulic and coalescence dominated regimes are well separated in time ($\bar{j}^a = 1$). Starting from the Gaussian profile used to generate the chain, the distribution first evolves toward the asymmetric distribution characteristic of hydraulic coarsening (Fig 4B). The transition toward the coalescence phase is characterized by a plateau for the number of lumens and a peaked distribution of lumen sizes, indicative of a collective growth phase with size homogenization of the lumens. This size homogenization can be understood by the fact that lumens become hydraulically uncoupled while coalescence has not yet started, leading to a narrowing of the distribution (see S1 Appendix). The coalescence-dominated regime exhibits then a multimodal distribution, that reveals subpopulations of lumens forming by successive waves of coalescence.

Finally, we study how spatial heterogeneities in active pumping within an embryo may bias the position of the final blastocoel. In mammalian embryos, the formation of the blastocoel relies on transepithelial active transport of ions and water from the external medium to the intercellular space by an apico-basally polarized cell layer, the trophectoderm (TE) [31, 57]. We expect therefore that active pumping may be essentially concentrated at cell interfaces with outer cells (TE), rather than at the ones between inner cells (called ICM for inner cell mass) as illustrated on Fig 6A. To study the effect of a spatial bias in active pumping, we consider a chain of $\mathcal{N}(0) = 100$ lumens and $\chi_v = 500$, and we perturb an uniform pumping profile with a

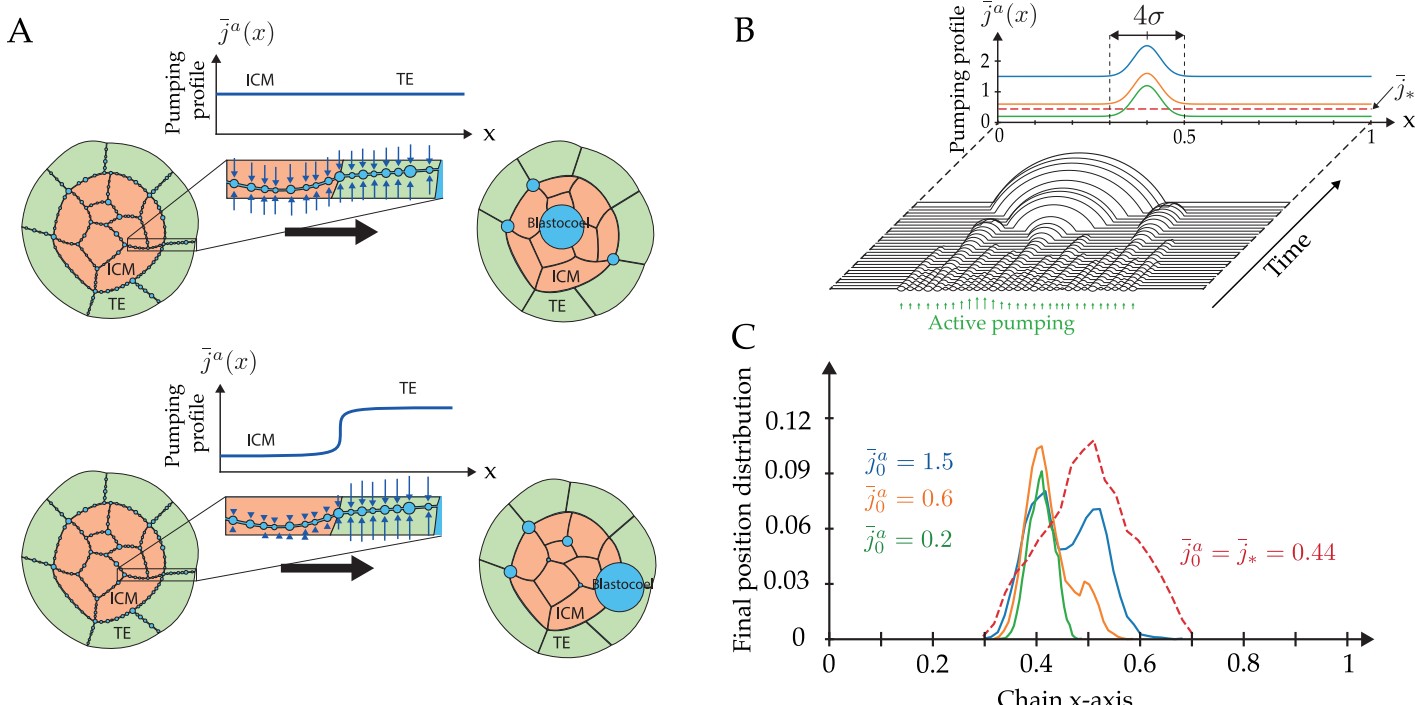

**Fig 6. Symmetry breaking from a spatial bias of active pumping.** (A) Schematic view of the coarsening outcome with and without spatially heterogeneous active pumping. This heterogeneity in pumping between TE and ICM cell-cell contacts can bias the positioning of the blastocoel toward the TE—ICM interface. (B) Plot of the pumping profiles along the chain and typical chain dynamics ($\mathcal{N}_0 = 30$). The uniform profile corresponds to the threshold $\bar{j}^* \simeq 0.44$ (dashed lines) and perturbed profiles are biased by a Gaussian in the form $\bar{j}^a(x) = \bar{j}_0^a + \frac{\delta\bar{j}^a}{\sqrt{2\pi}\sigma} \exp\left(-\frac{(x+\mu)^2}{\sigma^2}\right)$ with basal pumping rates $\bar{j}_0^a = 0.2, 0.6, 1.5$ and $\mu = 0.4$, $\sigma = 0.05$, $\delta\bar{j}^a = 1$. (C) Distributions for the localization of the final lumen on a chain of $\mathcal{N}_0 = 100$ lumens, corresponding to the pumping profiles depicted above; the red dashed curve corresponds to uniform profile ($\bar{j}^*$), the full lines to perturbed profiles ($\bar{j}_0^a = 0.2, 0.6, 1.5$). Each curve is obtained by averaging 10000 simulations with $\chi_v = 500$, $\chi_s = 1$.

Gaussian function $\bar{j}^a(x) = \bar{j}_0^a + \frac{\delta\bar{j}^a}{\sqrt{2\pi}\sigma} \exp\left(\frac{(x+\mu)^2}{\sigma^2}\right)$ that is shifted from the chain center ($\mu = 0.4$, $\sigma = 0.05$). We simulate the chain dynamics keeping the amplitude $\delta\bar{j}^a = 1$ of the Gaussian profile constant and changing the basal uniform value of pumping $\bar{j}_0^a$ only. A uniform pumping profile leads to a typical distribution centered at the middle of the chain. To evaluate the effect of the perturbation with respect to the uniform pumping, we calculate the area below a region of width $4\sigma$ centered on the Gaussian and compare it with the area below the remaining quasi-uniform part of the profile of value $\sim \bar{j}_0^a$ (see Fig 6B). This yields a typical threshold in basal pumping $\bar{j}_0^a = \bar{j}^* \equiv \delta\bar{j}^a \frac{\sigma\sqrt{\pi}}{0.6 \cdot 8\sigma} \text{erf}(2)$ (see S1 Appendix, Eq (58)). For a basal pumping rate $\bar{j}_0^a$ below the threshold $\bar{j}^* \simeq 0.44$, the perturbation dominates and the mean position of the final lumen is shifted toward the center $x = \mu$ of the Gaussian. This biasing of the final lumen position is in fact reminiscent of the reversed coarsening behavior found for two lumens (Fig 3A). On the contrary, when the basal pumping rate $\bar{j}_0^a > \bar{j}^*$, the two effects compete with each others, and the distribution for the final lumen localization may become bimodal (Fig 6C). In spite of rapid diffusion within the intercellular space, the spatial localization of active pumping can therefore break the radial embryo symmetry by positioning the blastocoel away from the center of the network. In addition to mechanical differences between TE and ICM cells [19, 58], different rates of pumping between these two types of cells may constitute an alternative and fail-safe mechanism to ensure the robust localization of the blastocoel at the TE interface of mammalian embryos.

## Discussion

We have presented a novel theory for the coarsening of a network of hydro-osmotically coupled biological cavities, that brings several generic, robust and experimentally testable predictions. From a physics perspective, coarsening processes have been the subject of a vast literature, with generally diffusion as the main mode of transport between dynamically coupled compartments [56]. Only a few of them have considered hydrodynamic flows as an alternative mean for mass exchange [53, 54] and the influence of osmotic effects in coarsening has been studied mostly as a mean to stabilize emulsions or foams [48, 49]. We find unexpectedly that our hydro-osmotic coarsening model exhibits, for a wide range of solvent and solute permeabilities, the same scaling dynamics as predicted for thin dewetting films, where an ensemble of droplets interacts through a thin fluid layer [47, 52, 55]. This indicates that water flows generated by osmotic heterogeneities have generically a mild influence on the coarsening behavior, that is ultimately dominated by hydraulic exchanges, as we hypothesized in [19]. Extrapolating our prediction $\mathcal{N} \sim t^{-2/5}$ for the number of lumens in a 1D chain to the more realistic topology of 3D microlumens forming a continuous 2D network at cell-cell interfaces, one expects a scaling law $\mathcal{N} \sim t^{-3/4}$ [54, 59, 60]. Whether the slightly more complex topology in embryo and tissues, where cell-cell contacts form themselves a 3D network, might affect the late scaling dynamics remains however to be explored. As this scaling behavior relies essentially on the hydraulic coupling between lumens, one needs to compare the typical diameter of an embryo or small tissue ($50 - 100\mu m$) with a typical pressure screening length $\xi_v \sim 84\mu m$ (see S1 Table). Since screening and embryo dimensions are of the same order, we expect lumens to remain hydraulically coupled during the whole coarsening process, hence ensuring the robust formation of a single cavity. To test and challenge this aspect, it could be interesting to artificially merge two or more mouse embryos before blastulation to see if several uncoupled cavities may form.

One major simplification in our theory is to have neglected the local mechanical response of surrounding cells to lumens deformations. As cell mechanics enters essentially in the small parameter $\epsilon = \frac{\gamma \sin \theta}{L_0 \Pi_0}$ in our model, we do not expect cell mechanics to fundamentally affect the scaling behavior on long timescales. We have hence verified that the scaling exponent remains unaffected by $\epsilon$, as long as it is smaller than $10^{-1}$ (see S4 Fig), which is larger than a typical upper estimation $\epsilon < 10^{-2}$ in embryos. Interestingly, in mouse embryos, shape oscillations of lumens (or cells) may be detected on short timescales of a few dozens of seconds [19], to compare with the hours duration of the coarsening process. In a similar model but for a single lumen only [36], a viscous contribution to cortical tension was added, and it was shown it could lead to spontaneous lumen shape oscillations. It could therefore be interesting to consider viscous surface deformations and possible emergence of short time-scale oscillations in our collective lumen context as well.

A second major assumption of our model is to have considered only a single osmolyte of zero charge, an approximation similar to [36, 61]. Yet, cell volume control is known to rely on the pump-leak mechanism, that involves, at minimum, three major ionic species $K^+$, $Na^+$ and $Cl^-$ [62–66]. Our choice was first motivated by the sake of simplicity but turned out to also largely ease the numerical solving of our equations. The total osmolarity difference between cytoplasmic and external compartments $\delta c$ remains generally small in animal cells compared to a typical absolute value $c_0$. This is what allowed us to calculate an analytic solution for the concentration in bridges, to reformulate the dynamics of the system as coupled ordinary equations and to perform extensive numerical simulations. On the contrary, each ion taken individually displays generally a large concentration difference between cytoplasmic and external compartments: $K^+$ is essentially cytoplasmic while $Na^+$ and $Cl^-$ are concentrated in the

external medium. No simplification of bridge equations would then be possible, and membrane potential gradients along the bridge would have to be explicitly considered, leading possibly to electrodiffusion or electroosmotic effects. These interesting extensions of our model are beyond the scope of this manuscript and will require ample effort in terms of theoretical modeling and numerical work. They open exciting novel avenues of research on the still largely overlooked roles of bioelectricity in development and tissue morphogenesis [67, 68].

In spite of these simplifications, we have shown importantly that active ion pumping can bias coarsening spatially. Active osmotic volume control emerges hence as a novel mode of symmetry breaking for tissue morphogenesis, that could be relevant also to oogenesis [37, 40, 41], to liquid phase transition of biological condensates [69] and to plant tissue growth [43, 45]. This symmetry breaking may complement or compete with mechanical gradients, that were shown to play a major role in mouse embryo blastocoel formation [19] or in the coarsening of membrane-less organelles [70, 71]. We have shown furthermore that active pumping can also largely change the collective lumen dynamics, with the emergence of a novel coarsening regime dominated by coalescence. If the effect of active pumping may be compared to droplet condensation on a surface [72, 73], the absence of nucleation in our model constitutes a major difference that precludes a direct comparison of early scaling behaviors [51, 74]. The inclusion of lumen nucleation would require a precise microscopic description of the de-adhesion process [75], coupling membrane mechanics to the stochastic binding-unbinding of adhesion molecules and to active volume control. The coupling between short-time scale mechanics and the long timescale coarsening dynamics are undoubtedly interesting avenues to further refine physical theories of lumenogenesis.

## Methods and materials

The model equations in the main text are derived in detail in the Appendices, along with additional studies.

Coupled differential equations are solved numerically using a Runge-Kutta-Fehlberg method (RKF45) with adaptive time-step [76, 77]. Topological transitions such as coalescence of two lumens or deletion of an empty lumen are automatically detected and handled by our numerical scheme (see S5 Fig and S1 Appendix for details). On Fig 3, the classification of events is determined by the sign of the final slopes for the time evolution of lumen area (see S1 Appendix for details).

The simulation code and Python notebooks for analysis of the results are available on the following repository: https://github.com/VirtualEmbryo/hydroosmotic_chain.

### 2-lumen Fate classification

Fate corresponds to the first event that occurs for a system of two lumens. An isolated lumen with small active pumping $\bar{j}^a < \epsilon/\bar{L}$ will collapse, otherwise it will grow until it occupies the whole chain. Fates are classified depending on the three last steps of the dynamics. The fates are classified as:

- Collapse: both lumens decrease in size. The three last steps of both lumen areas are decreasing.

- Coalescence: a *merge* event is detected, and a new lumen is created, hence stopping the simulation.

- Coarsening: one lumen shrinks, the other grows. If the initially smaller lumen is the one finally growing, the fate is considered as reversed coarsening.

### Scaling-law average

Figs 4C, 4D, 5A and 5B show time averages of tens of simulations. Because of different time-stepping from one simulation to another, we defined arbitrarily time points in log-scale at which the number of lumens $\mathcal{N}(t)$ are averaged within these predefined time windows.

### Size distributions

Figs 4B and 5C show size-distributions of lumens for $n \geq 100$ simulations. Distributions are calculated at a given simulation time step. Because of different time-stepping from one simulation to another, the closest time is chosen for each realization.

## Supporting information

**S1 Appendix. Details of the analytical model derivation and description of the numerical scheme.**
(PDF)

**S1 Table. Symbols, values and units.**
(PDF)

**S1 Fig. Net solvent flux diagrams.** Net solvent flux diagrams as function of the relative concentration asymmetry $\Delta_C$ and pressure asymmetry $\Delta_P$ for a 2-lumen system, at different values of the rescaled screening numbers $\chi_{s,v}$. Dashed lines represent the zero net flux $\bar{J}_{2 \to 1}^v = 0$. At large pressure screening number $\chi_v = 100$, the net flux does not depends on the concentration screening number $\chi_s$. For smaller pressure screening numbers $\chi_v = 0.01, 1$, we observe a mild influence of the concentration screening length when it approaches low values $\chi_s = 0.01$. This shows that the concentration screening length has limited influence on the net solvent fluxes compared to the pressure screening length. Asymmetry ratios are defined in the main text.
(EPS)

**S2 Fig. Half a lumen with moving boundary $X_\ell(t)$ for Stefan's problem (S1 Appendix, Section 1.3.3.).** The area of half the lumen is $A_{1/2}$ (light blue region) plus the bridge-lumen portion (dark blue region). The green region corresponds to the bridge area. Due to mass conservation and in the absence of permeation nor osmotic effects, the fluxes $J(t)$ and $J_B(t)$ are equal. The variation in lumen's area $A_{1/2}$ corresponds to outgoing flux $-J_B$, solving the moving boundary problem.
(EPS)

**S3 Fig. Typical dynamics of a chain with $\mathcal{N}(0) = 10$ lumens, with increasing solute relaxation time $\tau_s$.** (A) At low $\tau_s = 10^{-2}$ $s$ with respect to $\tau_v = 1s$, the dynamics of the chain is similar to the hydraulic chain, with little deviation of lumen's concentration from osmotic equilibrium ($\bar{c}_0 = 1$). (B) With increasing $\tau_s = 1s$, solutes are retained within lumens and larger concentration deviations are observed when they collapse. (C) At larger solute relaxation time $\tau_s = 10^2$ $s$, solutes are trapped within the lumens and intra-lumenal concentrations diverge when they collapse. In all cases, $\chi_v = 50$, $\chi_s = 5$, $\tau_v = 1s$ with the same initial conditions.
(EPS)

**S4 Fig. Influence of cell mechanics on coarsening.** The parameter $\epsilon$ accounting for cellular mechanics is varied from $10^{-4}$ to 1. (Left) Number of lumens as function of the rescaled time $t/\tau_v$, showing the influence of $\epsilon$ on the hydraulic time $T_h$. (Right) Number of lumens as function of the rescaled time $t/T_h$. All the curve collapse but a small deviation from the scaling law $t^{-2/5}$ is observed for $\epsilon \gtrsim 10^{-1}$, illustrating the role of Laplace pressure in the shrinkage phase of

lumens by speeding up lumen collapse and the mild influence of cellular mechanics in our model. The scaling law $\mathcal{N}(t) \sim t^{-2/5}$ is shown as a dashline. Parameters are $\chi_v = 50, \chi_s = 5, \tau_s = \tau_v = 1s$ and $\mathcal{N}(0) = 1000$. Each curve is an average of 20 simulations.
(EPS)

**S5 Fig. Representation of the numerical scheme.**
(EPS)

**S1 Video. Two-lumen dynamics with active pumping.** The four possible dynamics for 2 lumens, corresponding to the four cases highlighted on Fig 3A; collapse ($\chi_v = 6.8 \times 10^{-1}, \bar{j}_1^a = 3.2 \times 10^{-4}$, top left), coarsening ($\chi_v = 2.2 \times 10^{1}, \bar{j}_1^a = 3.2 \times 10^{-3}$, top right), reversed coarsening ($\chi_v = 6.8 \times 10^{-1}, \bar{j}_1^a = 3.2 \times 10^{-3}$, bottom left) and coalescence ($\chi_v = 6.8 \times 10^{-1}, \bar{j}_1^a = 3.2 \times 10^{-2}$, bottom right). Simulation time is shown on top for each simulation. Other model parameters are $\tau_s = \tau_v = 1s, \chi_s = 1$ and the lumens are at osmotic equilibrium at time $t = 0$.
(AVI)

**S2 Video. Coarsening dynamics of a chain of 30 lumens without pumping.** The simulation time is plotted on top. Due to adaptive time-step, rendering of the simulation "slows down" when a topological events happen. Parameters are $\bar{A}_0 = 1, \chi_v = 500, \chi_s = 5, \tau_s = \tau_v = 1s$. We chose $\bar{\ell}_0 = 1$ for the sake of illustration.
(AVI)

**S3 Video. Coarsening dynamics of a chain of 30 lumens with active pumping.** The simulation time is plotted on top. Labels indicate the different regimes of the chain hydraulic exchange, plateau, coalescence. Parameters are $\bar{A}_0 = 1, \chi_v = 100, \chi_s = 1, \tau_s = \tau_v = 1s, \bar{j}^a = 2$. We chose $\bar{\ell}_0 = 1$ for the sake of illustration.
(AVI)

## Acknowledgments

We thank Jean-Léon Maître's team for early discussions, Pierre Recho for useful comments on the manuscript, and the Turlier's team members for their support.

## Author Contributions

**Conceptualization:** Hervé Turlier.

**Data curation:** Mathieu Le Verge-Serandour.

**Formal analysis:** Mathieu Le Verge-Serandour, Hervé Turlier.

**Funding acquisition:** Hervé Turlier.

**Investigation:** Mathieu Le Verge-Serandour, Hervé Turlier.

**Methodology:** Mathieu Le Verge-Serandour, Hervé Turlier.

**Project administration:** Hervé Turlier.

**Resources:** Hervé Turlier.

**Software:** Mathieu Le Verge-Serandour.

**Supervision:** Hervé Turlier.

**Validation:** Hervé Turlier.

**Visualization:** Mathieu Le Verge-Serandour.

**Writing – original draft:** Mathieu Le Verge-Serandour, Hervé Turlier.

**Writing – review & editing:** Mathieu Le Verge-Serandour, Hervé Turlier.

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
