## [Decision Letter · Decision Letter 0]

29 Apr 2021

Dear Dr Turlier,

Thank you very much for submitting your manuscript "A hydro-osmotic coarsening theory of biological cavity formation" for consideration at PLOS Computational Biology.

As with all papers reviewed by the journal, your manuscript was reviewed by members of the editorial board and by several independent reviewers. In light of the reviews (below this email), we would like to invite the resubmission of a significantly-revised version that takes into account the reviewers' comments.

All of three reviewers have favorable comments on the work, and suggest it is a well-performed theoretical study that will contribute to our understanding on the dynamics of lumens. They provide some constructive comments on improving the presentation. Reviewer 2 also suggests some aspects to make the model more realistic. I suggest the authors to consider them carefully.

I understand one concern the authors has is existence of a competitive study likely under review somewhere. We will try to expedite the process. If you have any concern, please don't hesitate on contacting me or the journal office.

We cannot make any decision about publication until we have seen the revised manuscript and your response to the reviewers' comments. Your revised manuscript is also likely to be sent to reviewers for further evaluation.

Sincerely,

Jianhua Xing

Guest Editor

PLOS Computational Biology

Mark Alber

Deputy Editor

PLOS Computational Biology

All of three reviewers have favorable comments on the work, and suggest it is a well-performed theoretical study that will contribute to our understanding on the dynamics of lumens. They provide some constructive comments on improving the presentation. Reviewer 2 also suggests some aspects to make the model more realistic. I suggest the authors to consider them carefully.

I understand one concern the authors has is existence of a competitive study likely under review somewhere. We will try to expedite the process. If you have any concern, please don't hesitate on contacting me or the journal office.

Reviewer's Responses to Questions

**Comments to the Authors:**

Reviewer #1: This is a comprehensive theoretical and computational study of the collective dynamics of lumens, as driven by flow of water and osmolytes between cavities and by pumping of osmolytes. The authors predict whether coarsening in a single lumen occurs and discuss applicability of their results to experiments. The work and results are of potential high interest to developmental biologists, to biophysicists, and to applied mathematicians. It is definitely worth publishing, though the clarity of the writing needs to be improved.

CONTENTS

- What would change in a more realistic 3D geometry, with lumens of finite size and bridges in the form of tubes?

- L186. Why a factor of 2 before the sum of L_i?

- L225. What is the meaning of the timescale of 1s to which tau_s is compared? Normally one should compare tau_s to another timescale defined by problem parameters.

- How relevant is this work to other types of lumens?

WRITING

- The beginnings of the abstract and introduction are too technical. Please define (or refrain from using): apical, basolateral, blastocoel…

- The author summary should be written for a scientist that is far from the field.

- In caption of Fig. 1: A_i and mu may be omitted, otherwise they need to be defined

- L78. Why isn’t sin(theta) absorbed in mu? What is the definition of mu?

- L87. Perhaps the definition of L_i should appear earlier.

- P4. Briefly explain why bridge thickness is considered to be constant.

- P5. It is not well thought to call dimensionless numbers ’screening lengths’. Why not ‘screening numbers’?

- Eqs. 6-8, why mix dimensionless quantities and time? This may be misleading.

- In insets of Fig.3A, the differences in size between lumens 1 and 2 are too small to be visible.

- P7. A brief description of coalescence events is needed.

- L247. What does \\bar{L}_{ij} stand for?

Reviewer #2: I read this paper by Le Verge–Serandour and Turlier with interest. The problem of lumen formation is an interesting biophysical problem, with numerous unknown subtopics that require some detailed understanding. The treatment by the author is quite elegant mathematically, and provides useful simplifications for the physical problem. The paper is nicely done and generally clear, and can be published. However, I find some simplifications in the current form are unrealistic biologically. Since this is a journal about computational biology, the treatment can include more biological detail. In particular models of active ion flux exist. The paper can improve substantially with some revision and additions.

1) A major problem is, j^a or the ion flux is not a constant. j^a should depend on concentration differences, and energy input by cells (it's a pump), and potentially tension in the membrane. There are models of j^a already in the literature, for example, Jiang & Sun, Biophysical J. 2013, and more detailed treatment in subsequent papers with different ionic species.

2) The active water flux across an epithelial barrier has been measured, e.g., see Li et al, Journal of Cell Science, 2020. For constant lumen osmolarity, the water flux was found to be maximum at Delta P=0, and decreases with increasing Delta P. The reason for this is can be traced to the properties of j^a. It is true that a detailed theory for j^a is lacking, but there has been work on this. The expression of j^a as a function of ion concentration will change the results qualitatively. Can the authors add some results with more realistic j^a?

3) In the presented physics, the rate at which l decreases is solely due to lumen expansion (Eq. 8) from water inflow. But another importance piece seems to be the break down of tight junctions that keep the junction zipped. One can imagine a scenario where P builds up sufficiently so the tension increases, and therefore unzips the tight junction. The rate of unzipping may depend on various factors. Where does this physics come in?

4) In a biologically realistic lumen, is it true that gamma, gamma_c and therefore mu are constants spatially and temporally within the lumen? Does the tension change with time? I would think there is an active adjustment process. Is the assumption with constant theta realistic?

5) What happens when 2 lumens collide and l for the bridge goes to zero? It seems that some assumptions break down here. At this moment, L_1+L_2=1, and you still have 2 lumen, how does this transition to 1 lumen? I didn't find details about this. Right now there would be a jump in overall mass of the lumen (Fig. 3A). This jump doesn't happen in reality, and the new lumen is filled slowly. How does this change the result for N(t) in Fig. 4. Does this take some time, and therefore change the time scale?

Reviewer #3: The review is uploaded as an attachment

**Have the authors made all data and (if applicable) computational code underlying the findings in their manuscript fully available?**

Reviewer #1: Yes

Reviewer #2: Yes

Reviewer #3: Yes

PLOS authors have the option to publish the peer review history of their article (what does this mean?). If published, this will include your full peer review and any attached files.

Reviewer #1: No

Reviewer #2: No

Reviewer #3: No
---

## [Decision Letter · Decision Letter 1]

9 Aug 2021

Dear Dr Turlier,

We are pleased to inform you that your manuscript 'A hydro-osmotic coarsening theory of biological cavity formation' has been provisionally accepted for publication in PLOS Computational Biology.

Before your manuscript can be formally accepted you will need to complete some formatting changes, which you will receive in a follow up email. A member of our team will be in touch with a set of requests. Reviewer 3 listed a few minor change suggestions. I recommend you to consider and revise accordingly.

Best regards,

Jianhua Xing

Guest Editor

PLOS Computational Biology

Mark Alber

Deputy Editor

PLOS Computational Biology

Reviewer's Responses to Questions

**Comments to the Authors:**

Reviewer #1: The authors have addressed all the points that I raised.

Reviewer #2: The authors have answered my concerns. I recommend publication.

Reviewer #3: The review is attached as a pdf to this form.

**Have the authors made all data and (if applicable) computational code underlying the findings in their manuscript fully available?**

Reviewer #1: Yes

Reviewer #2: Yes

Reviewer #3: Yes

PLOS authors have the option to publish the peer review history of their article (what does this mean?). If published, this will include your full peer review and any attached files.

Reviewer #1: No

Reviewer #2: No

Reviewer #3: No

---

## [Editor Report · Acceptance letter]

25 Aug 2021

PCOMPBIOL-D-21-00531R1 

A hydro-osmotic coarsening theory of biological cavity formation

Dear Dr Turlier,

I am pleased to inform you that your manuscript has been formally accepted for publication in PLOS Computational Biology. Your manuscript is now with our production department and you will be notified of the publication date in due course.

With kind regards,

Zsofi Zombor
